# Low Energy Status under Methionine Restriction Is Essentially Independent of Proliferation or Cell Contact Inhibition

**DOI:** 10.3390/cells11030551

**Published:** 2022-02-04

**Authors:** Corinna Koderer, Werner Schmitz, Anna Chiara Wünsch, Julia Balint, Mohamed El-Mesery, Julian Manuel Volland, Stefan Hartmann, Christian Linz, Alexander Christian Kübler, Axel Seher

**Affiliations:** 1Department of Oral and Maxillofacial Plastic Surgery, University Hospital Wuerzburg, D-97070 Wuerzburg, Germany; corinna.koderer@posteo.de (C.K.); wuenschanna@outlook.de (A.C.W.); julia-balint@web.de (J.B.); julian_volland@gmx.de (J.M.V.); hartmann_s2@ukw.de (S.H.); linz_c@ukw.de (C.L.); kuebler_a@ukw.de (A.C.K.); 2Department of Biochemistry and Molecular Biology, Biocenter, D-97074 Wuerzburg, Germany; wschmitz@biozentrum.uni-wuerzburg.de; 3Department of Biochemistry, Faculty of Pharmacy, Mansoura University, Mansoura 35516, Egypt; m_elmesery@mans.edu.eg

**Keywords:** methionine restriction, caloric restriction, mass spectrometry, LC/MS, liquid chromatography/mass spectrometry, metabolomics, L929, amino acid, proliferation, contact inhibition

## Abstract

Nonlimited proliferation is one of the most striking features of neoplastic cells. The basis of cell division is the sufficient presence of mass (amino acids) and energy (ATP and NADH). A sophisticated intracellular network permanently measures the mass and energy levels. Thus, in vivo restrictions in the form of amino acid, protein, or caloric restrictions strongly affect absolute lifespan and age-associated diseases such as cancer. The induction of permanent low energy metabolism (LEM) is essential in this process. The murine cell line L929 responds to methionine restriction (MetR) for a short time period with LEM at the metabolic level defined by a characteristic fingerprint consisting of the molecules acetoacetate, creatine, spermidine, GSSG, UDP-glucose, pantothenate, and ATP. Here, we used mass spectrometry (LC/MS) to investigate the influence of proliferation and contact inhibition on the energy status of cells. Interestingly, the energy status was essentially independent of proliferation or contact inhibition. LC/MS analyses showed that in full medium, the cells maintain active and energetic metabolism for optional proliferation. In contrast, MetR induced LEM independently of proliferation or contact inhibition. These results are important for cell behaviour under MetR and for the optional application of restrictions in cancer therapy.

## 1. Introduction

A late event in the development of cancer is unlimited proliferation and the resulting space-occupying lesion. Although the pathological causes of carcinogenesis are diverse [1] and proliferation is an important biological process regulated and influenced by numerous factors [2,3,4], this process can be reduced to two simple regulating components: the availability of energy and mass. Both factors are essential, and proliferation can be fundamentally regulated via both elements. In the case of mass, amino acids are paramount. These molecules form the main mass of the cell [5] as structural proteins, enzymes, and numerous metabolites. In the case of energy, carbohydrates and lipids are more important initially and essentially result in the energy currencies ATP and NAD(P)H.

Continuous limitation of the mass via protein (PR) or amino acid restriction (AR) or in the case of energy by caloric restriction (CR) has a strong positive effect on almost all organisms. The absolute/relative lifespan is significantly extended, and the risk of age-associated cardiovascular disease, type II diabetes, and cancer is substantially reduced [6,7,8]. Therefore, common to all these forms of restriction is the induction of “low energy metabolism” (LEM), which manifests itself at the cellular level mainly in two biological responses: inhibition of proliferation and induction of autophagy. Both processes are important for resource conservation and recycling [9]. The implementation of the various forms of restriction has already been well elucidated at the molecular level. In general terms, a cell is either growing/dividing or not. Many sensors enable the cell to continuously measure the levels of energy and necessary building blocks and to determine whether cell division is possible and energetically sensible. With the help of a sophisticated molecular network, both intracellular and extracellular signals can be permanently recorded and processed. The AMP/ATP ratio, for example, is measured by AMP kinase (AMPK) [10]; the NAD^+^ level via sirtuins [11]; and the content of selected amino acids (e.g., leucine, arginine, glutamine, serine, and methionine) via various protein complexes, such as SAMTOR, which can be used to indirectly measure the methionine content via the intermediate S-adenosylmethionine (SAM) [12]. A deficiency of one of the necessary resources leads to the activation of different proteins and protein complexes, which in turn can have an inhibitory effect on one of the central switching sites, mechanistic target of rapamycin (mTOR). The sum of signals converging on mTOR determines whether mTOR actively promotes proliferation/growth or whether the cell switches to LEM by inhibiting mTOR and activating autophagy, among other things [13,14,15]. Extracellular signals from growth factors, such as the growth hormone (GH)-insulin-like-growth factor (IGF1) axis, also play a role here. IGF1 induced by GH and mostly secreted by the liver, mTOR, for example, is activated intracellularly via the PI3K/Akt pathway via the IGF1 receptor [16].

One of the simplest and most effective forms of restriction is methionine restriction (MetR), which is very easy to implement in cell culture by simply removing the amino acid in the medium. Many of the mechanisms induced by MetR are consistent with AR, PR, and CR [17,18,19]. In a previous work, we demonstrated that the murine cell line L929 reacts rapidly and efficiently to MetR. Proliferation was inhibited after 24 h, and the analysis over a period of 5 days of more than 150 different metabolites belonging to different classes [amino acids, urea and tricarboxylic acid cycle (TCA) cycles, carbohydrates, etc.] by liquid chromatography/mass spectrometry (LC/MS) defines a metabolic fingerprint and enables the identification of specific metabolites representing normal or MetR conditions. In addition to the large fingerprint with numerous metabolites, the induction of LEM can potentially be analysed using a small footprint with selected metabolites that are specific/characteristic of MetR-induced LEM and are composed of the combination of acetoacetate, creatine, spermidine, GSSG, UDP-glucose, pantothenate, and ATP [20].

Cells in culture usually have a similar metabolism to cancer cells. The reason lies in the proliferation itself, which leads to a change in metabolism called the Warburg effect. Thus, the Warburg effect is not a specific characteristic of cancer cells but of proliferating cells [4]. However, in contrast to tumour cells, many cells in culture retain the native capacity for contact inhibition, i.e., that adjacencies, including those caused by cell-cell contacts, lead to inhibition of proliferation [21,22,23].

In this work, we investigated the extent to which contact inhibition affects the metabolism of L929 cells and whether inhibition of proliferation induces a metabolic profile equivalent to that of LEM. For comparison, we cultured L929 cells under the same conditions under MetR. Over a period of 5 days, LC/MS was used to profile more than 150 metabolites every 24 h. In addition, the profiles were compared with the metabolic profiles of cells under proliferative conditions from the previous work mentioned above.

Interestingly, the cells retain their characteristic profile largely independent of proliferation. In principle, proliferation and contact inhibition have a slight influence on the metabolic profile, but the determining factor is whether it is full medium or MetR. Under full medium, cells show a tendency to maintain active and energetic metabolism to be prepared for optional proliferation at any time. In contrast, MetR induces a metabolism equivalent to that of LEM under both proliferative and confluent conditions. These results are important, as they demonstrate the possibility of inducing cells to develop LEM in principle. This phenomenon can be important in cancer therapy, among other applications.

## 2. Materials and Methods

### 2.1. Cell Culture

The murine fibroblast cell line L929 was purchased from the Leibniz Institute, DSMZ-German Collection of Microorganisms and Cell Cultures GmbH (Braunschweig, Germany). The parent L strain was derived from normal subcutaneous areolar and adipose tissue of a 100-day-old male C3H/An mouse. NCTC clone 929 (Connective tissue, mouse) of strain L was derived in March, 1948. Strain L was one of the first cell strains to be established in continuous culture, and clone 929 was the first cloned strain developed. Clone 929 was established (by the capillary technique for single cell isolation) from the 95th subculture generation of the parent strain (information from the homepage of ATCC—American Type Culture Collection: https://www.atcc.org/products/ccl-1 (accessed on 2 January 2022)). The cells were cultured in RPMI 1640 medium (Gibco, Life Technologies; Darmstadt, Germany) with 10% FCS (Sigma-Aldrich, Darmstadt, Germany) and 1% penicillin/streptomycin (P/S; 100 U/mL penicillin and 100 µg/mL streptomycin, Thermo Fisher Scientific, Darmstadt, Germany) at 37 °C in a humidified atmosphere containing 5% CO_2_. The basis medium lacked methionine. For full medium, 15 mg/L methionine (Sigma-Aldrich, Darmstadt, Germany) was added.

### 2.2. ImageXpress Pico Automated Cell Imaging System—Digital Microscopy (Pico Assay)

Cells were seeded at 10,000 cells in 100 µL of culture medium per well of a 96-well plate and incubated overnight. The following day, the cells were incubated in complete or methionine-free media. The incubation time is stated in the corresponding figure legend. For staining, 10 µL of Hoechst staining solution [1:200 dilution in medium of Hoechst 33342 (Thermo Fisher, Darmstadt, Germany) (10 mg/mL in H_2_O)] was added to each well, and the samples were analysed after a 20–30 min incubation. Wells were analysed with an ImageXpress Pico Automated Cell Imaging System (Molecular Devices, San Jose, CA, USA) via automated digital microscopy. The cells were analysed with transmitted light and in the DAPI channel at 4× magnification. The complete area of every well was screened. Focus and exposure time were set via autosetup and controlled by analysing three to four test wells. Finally, every result was confirmed visually, and 95% or more of cells were counted and analysed.

### 2.3. L929 Experiments for LC/MS

L929 cells were seeded in 20 mL of medium in 15 cm Petri dishes, and every value was measured in triplicate. For cell confluence, 3 × 10^6^ cells were seeded, and after 48 h, cells were stimulated with complete or Met(−) media. In an earlier work, 1 × 10^6^ cells/Petri dish were seeded under proliferative conditions for days 1, 2, and 3, and 5 × 10^5^ cells were seeded for days 4 and 5 to prevent confluence during the test period [20]. The media used for stimulation were prepared from methionine-free RPMI medium. The complete medium (control) contained 15 mg/L methionine, and the Met(−) medium lacked methionine (amino acid from Sigma-Aldrich, Darmstadt, Germany). All media contained 10% FCS (Sigma-Aldrich, Darmstadt, Germany) and 1% P/S (100 U/mL penicillin and 100 µg/mL streptomycin (Thermo Fisher Scientific, Darmstadt, Germany)). After seeding, the cells were incubated with 20 mL of complete medium or 20 mL of methionine-free medium per dish. Before harvesting, 1 mL of the supernatant was stored for analysis. The remaining medium was then removed, and the cells were washed with 10 mL of PBS and detached with 3 mL of trypsin/EDTA (Thermo Fisher Scientific, Darmstadt, Germany). After the addition of 7 mL of the appropriate medium, the absolute cell number in the suspensions was analysed with the automated cell counter EVE^TM^ (NanoEntek (VWR, Darmstadt, Germany)). Each sample was measured four times, and the mean value was calculated to obtain an accurate result. Pellets with 1 × 10^6^ cells were produced by centrifugation (5 min at 1200 rpm at RT). Until the LC/MS analysis, all samples were stored at −20 °C.

### 2.4. LC/MS

Analysis of water-soluble metabolites in cell extracts and culture media.

Cells: After the addition of 0.5 mL of MeOH/CH_3_CN/H_2_O (50/30/20, *v*/*v*/*v*) containing 10 µM lamivudine, cell pellets were homogenized by ultrasound treatment (10 × 1 s, 250 W output energy). Media: One hundred microliters of culture medium was combined with 0.4 mL of MeOH/CH_3_CN (50/30, *v*/*v*) containing 10 µM lamivudine. The external standard lamivudine was not used for absolute metabolite quantification but was used as a quality control to compensate for eventually occurring technical issues. As quality control and for the determination of the corresponding retention times, most of the annotated metabolites (which are commercially available) were run as mixtures of pure compounds under identical experimental conditions. General procedure: The resulting suspension was centrifuged (20 kRCF for 2 min in an Eppendorf centrifuge 5424), and the supernatant was applied to a C18-SPE column that was activated with 0.5 mL of CH_3_CN and equilibrated with 0.5 mL of MeOH/CH_3_CN/H_2_O (50/30/20, *v*/*v*/*v*). The SPE eluate was evaporated in a vacuum concentrator. The resulting pellet was dissolved in 50 µL (cell extracts) or 500 µL (media extracts) of 5 mM NH_4_OAc in CH_3_CN/H_2_O (25%/75%, *v*/*v*).

LC parameters: Mobile phase A consisted of 5 mM NH_4_OAc in CH_3_CN/H_2_O (5/95, *v*/*v*), and mobile phase B consisted of 5 mM NH_4_OAc in CH_3_CN/H_2_O (95/5, *v*/*v*).

After the application of 3 µL of the sample to a ZIC-HILIC column (at 30 °C), the LC gradient program was as follows: 100% solvent B for 2 min, a linear decrease to 40% solvent B over 16 min, maintenance at 40% solvent B for 9 min, and an increase to 100% solvent B over 1 min. The column was maintained at 100% solvent B for 5 min for column equilibration before each injection. The flow rate was maintained at 200 μL/min. The eluent was directed to the ESI source of the QE-MS from 1.85 min to 20.0 min after sample injection.

The MS parameters were as follows: scan type, full MS in the positive-and-negative mode (alternating); scan range, 69–1000 *m*/*z*; resolution, 70,000; AGC-target, 3E6; maximum injection time, 200 ms; sheath gas, 30; auxiliary gas, 10; sweep gas, 3; spray voltage, 3.6 kV (positive mode) or 2.5 kV (negative mode); capillary temperature, 320 °C; S-lens RF level, 55.0; and auxiliary gas heater temperature, 120 °C. Annotation and data evaluation: Peaks corresponding to the calculated monoisotopic masses (MIM +/− H^+^ ± 2 mMU) were integrated using TraceFinder software (Thermo Scientific, Bremen, Germany). Materials: Ultrapure water was obtained from a Millipore water purification system (Milli-Q Merck Millipore, Darmstadt, Germany). HPLC–MS solvents, LC–MS NH_4_OAc, and lamivudine were purchased from Merck (Darmstadt, Germany). The RP18-SPE columns were 50 mg of Strata C18-E (55 µm) in 1-mL tubes (Phenomenex, Aschaffenburg, Germany). The sonifier was a Branson Ultrasonics 250 equipped with a 13-mm sonotrode (Thermo Scientific, Bremen, Germany).

LC/MS system: A Thermo Scientific Dionex UltiMate 3000 UHPLC system linked to a Q Exactive mass spectrometer (QE-MS) equipped with a HESI probe (Thermo Scientific, Bremen, Germany) was used. The samples were analysed with a high-resolution mass spectrometer, allowing the generation of XIC data that were analysed by applying a very narrow *m*/*z* margin (+/− 3 mMU). The particle filter was a Javelin filter with an ID of 2.1 mm (Thermo Scientific, Bremen, Germany). The UPLC-precolumn was a SeQuant ZIC-HILIC column (5-μm particles, 20 × 2 mm) (Merck, Darmstadt, Germany). The UPLC column was a SeQuant ZIC-HILIC column (3.5-μm particles, 100 × 2.1 mm) (Merck, Darmstadt, Germany).

Raw Data Analysis and Value Generation (in short):

LC/MS analyses were carried out in four independent experiments at 24, 48, 72, 96 and 120 h, with each value obtained from triplicate measurements. Metabolites were quantified in cell pellets and corresponding supernatants (media) under methionine-supplemented and methionine-free conditions (12 samples per time point in total). The resulting peak areas were normalized against that of lamivudine as an external standard. From this, the mean value and standard deviation were calculated for each triplicate. For better comparisons, the values were converted to percentages. For the values of the media, the control measurement of the medium used was defined as 100%. For the cell pellets, the highest measured value in each test series within an experiment was defined as 100%. From these values, the average mean values from the four experiments were then summarized in the individual tables. For a better overview, the results were rounded to natural numbers and shown as a heatmap. The corresponding colour range is indicated individually under each table. The raw data and results for the two profiles are added as excel files in the Appendix A (proliferative profile) and Appendix A (non-proliferative profile).

### 2.5. Statistical Analysis

Data collection and plotting were performed with Excel (Microsoft, Redmond, WA, USA) and GraphPad Prism (version 6.04; GraphPad Software, San Diego, CA, USA) software. Statistical analysis was performed using GraphPad. Comparisons between different groups were performed by applying one-way ANOVA followed by the Tukey–Kramer multiple comparison test (*** *p* < 0.001).

## 3. Results

In a previous work, we established the murine cell line L929 as a model system to analyse MetR [20]. Thus, we deliberately chose a murine cell line for several reasons. First, much of the research on different restriction forms has been studied in rodent models. The mouse has established itself as a successful model system to analyse energy metabolism in this context [24]. The use of a murine system thus allows a better comparison with results from the literature. Second, the murine metabolism is considerably faster and more efficient. If one compares the metabolic equivalents energy/body weight/time (Kcal/g/h), the mouse has a metabolic turnover up to 100× higher than humans [25,26,27]. For this reason, murine cells are much better suited for analyses of the metabolome than human cells. A third reason is that a large number of mechanisms are strongly conserved or are implemented in a very similar way. With the beginning of the first cell, there was a great selection in the areas of energy and mass, which led to a large number of basic mechanisms. This is why, as already mentioned, the restrictions work in the most organisms, from yeast to nematodes, from Drosophila and rodents to primates and even humans [6,8]. Many conserved mechanisms are used, which, although adapted in the individual species, are nevertheless highly conserved. mTOR and the sirtuins are just two examples [11,14]. The murine cell line L929 has fulfilled these conditions. In addition, this cell line corresponds to fibroblasts, which are characterised by a general profile in contrast to neurons or hepatocytes, which are highly specialised.

As previously reported, proliferation was almost completely inhibited in L929 after 24 h. Importantly, over the study period of 120 h, even complete MetR (0 mg/mL) did not lead to a significant decrease in cell number, e.g., due to cell death (Figure 1a). In addition, the cells became increasingly sensitive to a decreasing methionine level over time (Figure 1b).

To determine how contact inhibition under full medium and under MetR affects metabolism, we seeded an appropriate number of cells on cell culture dishes and incubated them for 48 h before stimulation until the cells formed a confluent cell layer. Then, the cells were incubated with full medium or Met(−) medium (0 mg/mL) for 5 days each, and the metabolic profile was analysed every 24 h by LC/MS. The experiment was performed four times. The results of the four experiments are summarized in the following heatmaps. For clarity, only selected results are presented. The overall results are shown in the Appendix A for all experimental series. In addition, the results were compared with the results of a previous work in which the experiments were performed under proliferative conditions [20].

### 3.1. Under Full Medium, the Cells Replenish the Pool with All Amino Acids, but under MetR, Only Selected Amino Acids Are Added

In an analysis of the intracellular content of proteinogenic amino acids in full medium, for almost all amino acids, the highest concentration of 100% was reached after 120 h (Figure 2a). Basically, the cells replenish the amino acid pool despite contact inhibition. Even for amino acids that do not reach the maximum level (e.g., isoleucine and lysine), the level is kept as high as possible. The only exception was aspartate, and the value was reduced from 96% after 48 h to 61%. This finding contrasts with the profile under MetR. Here, the values partly decreased (e.g., asparagine and glycine) or were kept at a low level (e.g., cysteine and threonine). However, individual amino acids may have a special role. In addition to aspartate, which was strongly consumed under full medium conditions, and arginine, which also had a high level under full medium, the amino acids lysine and tyrosine accumulated at maximum levels under MetR.

In a comparison of the intracellular profile under confluent conditions with that under proliferative conditions (Figure 2b), the common tendencies are striking and are almost identical over large areas. Under full medium, the cells refuelled the amino acid pool, whereas under MetR, the cells largely abstained from amino acid uptake. However, under both proliferative and confluent conditions, the amino acids arginine, lysine, and tyrosine were more important, as they were basically present at very high levels, than the other amino acids and were even taken up by the cells. These amino acids may play a special role in the LEM under MetR.

Similar observations were found when comparing the profiles of extracellular concentrations in the medium. Under full medium, the amino acid content in the medium decreased for the majority of amino acids, while it increased under MetR for many amino acids. Although the cells under MetR still took up amino acids at the beginning, which was especially obvious after 24 h under confluent conditions, as time increased, many amino acids remained at a relatively constant level, indicating a stagnant or balanced amino acid import. The obvious exceptions were some amino acids that are strongly secreted (e.g., glycine); however, these amino acids were secreted under all conditions and thus likely have important functions in general, independent of proliferation and methionine content. Only the concentrations varied in the different conditions. As an example, glutamate was secreted to a fairly similar extent under all conditions. Alanine had a different pattern. While alanine was strongly secreted under the other conditions, the content decreased to a low value (23%) under confluent conditions in the full medium.

### 3.2. Substance Classes and Metabolic Pathways

In addition to the importance of the relative concentration of individual molecules, the summed relative masses of all relevant metabolites belonging to one pathway (e.g., the urea cycle) provide relevant information about its overall regulation. This type of analysis was performed for amino acids (except methionine), the urea cycle, the TCA cycle, carbohydrates, pyrimidines, and purines in both media and cell extracts under confluent and proliferative conditions (Figure 3a–d).

For the group of extracellular amino acids, the results already described can also be shown in total (Figure 3a,b). While the amino acid content decreased under full medium, it increased, albeit only slightly under MetR, under both confluent and proliferative conditions. The metabolic pathways of urea and the TCA cycle and the classes of pyrimidines and purines showed similar trends under all extracellular conditions and differed only in intensity. In contrast, the class of carbohydrates is an exception. Under proliferative conditions, the secreted amounts increased strongly, whereas under confluent conditions, probably also due to the much higher cell number, both under full medium and MetR, the sum of metabolites decreased drastically at the beginning. Under MetR, however, the trend towards increased secretion was again observed; under confluent conditions, the secretion in the full medium slowly stagnated and even decreased further to below the initial value after 24 h at the end.

Intracellularly, the sums of amino acids were consistent with the previous results. While the concentration increased under full medium and was maintained at a high level, the values decreased under MetR or remained at a constant level under both confluent and proliferative conditions. All other groups showed heterogeneous results overall. Thus, metabolites remain at rather high levels under full medium and increase under proliferative conditions, whereas they tend to decrease under MetR. Similar results were found for the TCA cycle. For the groups of carbohydrates, pyrimidines, and purines, the levels decreased under proliferative conditions and MetR, whereas the decrease was less pronounced or remained constant under confluent conditions. Under full medium, the opposite pattern was observed. Here, the concentrations in the three groups mentioned above increased or remained constant under proliferative conditions until the end of the experiment. The carbohydrates showed a cyclic pattern, always showing a temporal decrease followed by an increase.

### 3.3. MetR Induces LEM in L929 Cells at the Metabolic Level and Is Essentially Independent of Proliferation and Contact Inhibition

That the LEM is a program that is more or less independent of proliferation or contact inhibition is particularly evident from the so-called metabolic footprint, a selected group of metabolites. In a previous work, we defined the metabolic footprint as specific to the LEM under MetR, as these products either reflected appropriate biological activity and/or exhibited extreme differences under the conditions. The selected molecules were ATP, acetoacetate, creatine, spermidine, GSSG (glutathione in the oxidized form), UDP-glucose, and pantothenate. Thus, ATP is an ideal indicator of energy levels due to its biological function. Spermidine, for example, indicates high energy metabolism in cells when present at high concentrations and can inhibit proliferation when present at low concentrations or induce autophagy when taken up extracellularly, thus reflecting important indicators of restriction by its concentration [28,29]. The footprint should be used in studies to identify LEM as quickly as possible, even with alternative measurement methods (ELISA, Western blot, etc.), and is thus independent of costly mass spectrometry. Therefore, caloric restriction mimetics (e.g., metformin or rapamycin) [30,31] can be analysed in L929 cells or agents potentially capable of inducing CR via alternative pathways.

For this reason, the analysis of the metabolome under confluent conditions is also a quality control of whether the metabolite group we defined is suitable as a marker. When we compared the footprint under confluent and proliferative conditions (Figure 4a,b), we found that the metabolic profiles tended to be almost identical in a broad range of areas and are characteristic of MetR and full medium, respectively. The ATP content sharply decreased under MetR at a very early time point, whereas the concentration of acetoacetate increased continuously. The percentage trend for creatine was almost identical, increasing continuously in full medium and decreasing continuously under MetR, regardless of proliferation or confluence. Spermidine, as an activity marker, remained at a very high level under full medium, whereas due to the lower energy balance under MetR, spermidine content continuously decreased. GSSG, the oxidized form of glutathione, also increased independently of proliferation in full medium, while under MetR, the concentration remained low. The opposite result was found for UDP-glucose. Finally, panthotenate showed a constantly high level under full medium but a significantly lower content under MetR.

## 4. Discussion

In general, the different forms of restriction in both cells and organisms show many similarities at the molecular and metabolic levels, which usually lead to an extension of the lifespan over a long period of time as well as to the prevention of classic age-associated diseases such as type II diabetes, cardiovascular diseases, and cancer. MetR is a suitable method to generally investigate the effects and relevance of amino acid restriction in cells and organisms. In a previous work, we showed that MetR leads to the induction of LEM in the murine cell line L929 at the metabolic level. We used LC/MS to define both a comprehensive fingerprint and a footprint comprising only a few metabolites, which is sufficient to analyse and define MetR.

In this work, we investigated the extent to which contact inhibition affects the metabolism of L929 cells and whether inhibition of proliferation induces a metabolic profile equivalent to that of an LEM. We cultured L929 cells under confluent conditions using full medium and MetR for a period of 5 days and analysed the profile of more than 150 metabolites every 24 h by LC/MS. In addition, the profiles were compared with the metabolic profiles of cells under proliferative conditions from the previous work mentioned above [20].

Interestingly, the cells retained their characteristic profile independent of proliferation or confluence. In principle, proliferation and contact inhibition have a slight influence on the metabolic profile, but the determining factor is whether it is full medium or MetR. Under full medium, cells showed a tendency to maintain active and energetic metabolism to be ready for optional proliferation at any time point. In contrast, MetR induced metabolism equivalent to that of an LEM under both proliferative and confluent conditions. This phenomenon is particularly evident from the comparison of footprints under confluent and proliferative conditions, which are nearly identical (Figure 4a,b). These results are important, as they demonstrate the possibility of inducing cells to develop LEM in principle and at any time point. In general, chemical systems are very complex and can be influenced by a multitude of external factors that lead to special and situation-specific reactions under stress conditions [32]. However, we assume that the LEM is an intrinsic program that is already established in unicellular organisms and strongly conserved in evolution. The cross-species involvement of mTOR and the sirtuins in the regulation of LEM are examples [11,14].

This finding may be important for cancer therapy. Longo and colleagues showed in a series of studies that prolonged restriction is necessary, but that short-term fasting (no calorie intake) or fasting-mimicking diets (FMDs) lead to varied alterations in growth factors and in metabolite levels, generating environments that can reduce the ability of cancer cells to adapt and survive, and thus improving the effects of cancer therapies. In addition, fasting or FMDs increased resistance to chemotherapy in normal but not cancer cells and promoted regeneration in normal tissues, which could help prevent detrimental and potentially life-threatening side effects of treatments [33]. This result is in extreme contrast to frequent recommendations, e.g., by the American Cancer Society, to increase calorie and protein intake in cancer patients [34]. In mammals, as well as humans, fasting leads to cell protection. The protective effect is mediated by, among other things, a strong reduction in the growth factor IGF-1. In addition, fasting-induced proto-oncogenes act as important negative regulators. This result further leads to an extreme situation called “differential stress resistance”. Cancer cells expressing oncogenes themselves and with egocentric proliferative behaviour essentially respond to selected cancer-promoting and growth factors. Thus, cancer cells also do not respond to the protective signals generated by fasting in the short term or by restrictions in the long term. For this reason, the cells are in two extreme situations: somatic cells are protected, while cancer cells become additionally vulnerable to attack [35]. This phenomenon is especially true when cancer cells do not limit proliferation due to restriction but maintain their high energy and thus proliferative mode.

A good example is the use of cisplatin in combination with fasting. Short-term starvation (STS) based on calorie and/or protein reduction protects normal cells while simultaneously sensitizing malignant cells to high-dose chemotherapeutic drugs such as cisplatin in mice and possibly patients. The fasting-dependent protection of normal cells and sensitization of malignant cells depend, in part, on reduced levels of IGF-1 and glucose [36]. However, success against cancer is tumour dependent. While glioma progression could not be delayed in mouse models [36], work on cisplatin-resistant non-small human cell lung cancer and ovarian cancer cell lines showed that cisplatin-resistant clones were more sensitive to killing by nutrient deprivation in vitro and in vivo. Mass spectrometry revealed glutamine as central for nucleotide biosynthesis rather than for anaplerotic and bioenergetic reactions in cisplatin-resistant cells. Glutamine depletion was sufficient to restore the cisplatin responses of initially cisplatin-resistant cells [30,37].

Additionally, in this work, it was shown under both proliferative and confluent conditions that amino acids represent an ideal target, which is why they have become increasingly important in tumour therapy over the last decade [38,39]. The central role is based, among other things, on the fact that the essential mass of proliferating cells is generated from extracellularly absorbed amino acids [5]. As already mentioned, glutamine, among others, plays an important role here, as glutamine is the extracellular transport amino acid par excellence and can thus replenish the citrate cycle intracellularly distributed via the blood by means of glutaminolysis but also plays a role in other biological processes, such as nucleotide synthesis [40].

Based on the results of LC/MS under proliferative and confluent conditions in MetR, two relevant aspects are notable. A large portion of the amino acids are barely taken up intracellularly or even secreted into the medium, thus playing only a minor role in the LEM status of the cell. Other amino acids continue to be absorbed or stored to a greater extent under MetR. However, the question arises to what extent the cell truly needs these in the LEM or whether a reserve is not simply created for better, i.e., highly proliferative conditions. Regardless of the loading status of the amino acids, under LEM conditions, they are only needed to a much lesser extent or at a later time point. These results are in contrast to the tumour, reflected in part in this work by the full medium conditions, which permanently tries to remain in high energy mode and is thus prepared for optional proliferation.

It is precisely for this reason that the tumour is extremely vulnerable to attack. If the amino acid supply is inhibited, e.g., by using an amino acid transport inhibitor or an inhibitor of glutaminolysis, this will essentially affect the tumour but not the frugal and nonproliferating cells in LEM mode. A possible candidate would be V9302, which is also already successfully used in clinical trials, and glutaminolysis inhibitors [41,42].

How effective protein restriction can be in preventing tumours has already been shown by Levine and colleagues, both in a human study and in in vivo experiments in mouse models [43]. How successful would a combination of restriction and various amino acid inhibitors be? It remains to be noted that the induction of an LEM under MetR at the metabolic level is not significantly affected by proliferation or contact inhibition and that the footprint we defined reflects the energetic state under all conditions in L929.

Finally, the results demonstrate that by means of MetR, a low energy metabolism is induced independent of the status of proliferation, which differs substantially from the active and proliferation-oriented metabolism of a (tumour) cell. Our results are of additional importance within tumour therapy, as they show the possibility to further differentiate the metabolism of tumour cells from healthy cells by means of MetR and thus significantly expand the therapy options.

## Figures and Tables

**Figure 1 cells-11-00551-f001:**
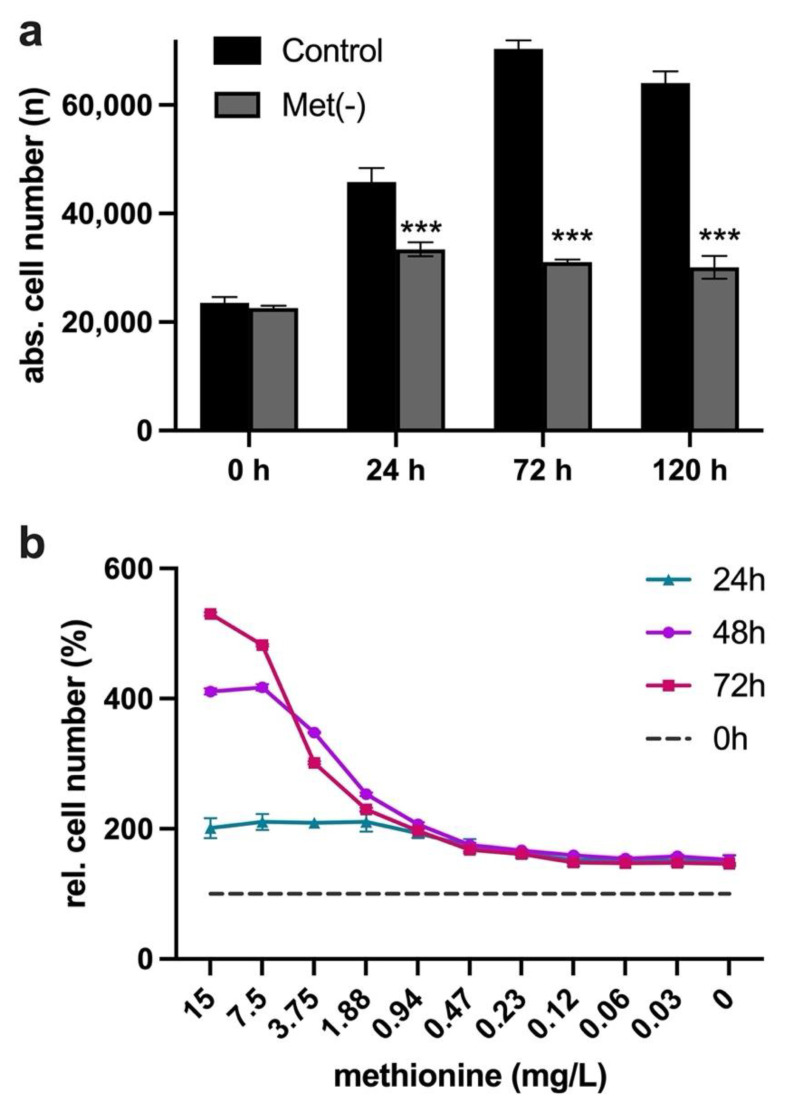
(**a**,**b**) Analysis of L929 cell proliferation under MetR. A total of 10,000 cells were seeded per well and incubated overnight. (**a**) Then, the cells were stimulated for 24, 72 and 120 h with or without methionine. The proliferation of the cells was analysed via ImageXpress digital microscopy analysis as described in the Materials and Methods. The figure shows one representative experiment (five values for every group). (**b**) The cells were incubated for 24, 48 and 72 h with a log2 dilution of methionine. The proliferation of the cells was analysed via ImageXpress digital microscopy analysis as described in the Materials and Methods. The figures show a summary of the results obtained from three independent experiments (two values for every group per experiment). Statistical analysis was performed using GraphPad Prism 5.0. Comparisons between Control and Met(−) groups were performed by applying one-way ANOVA followed by the Tukey–Kramer multiple comparison test (*** *p* < 0.001).

**Figure 2 cells-11-00551-f002:**
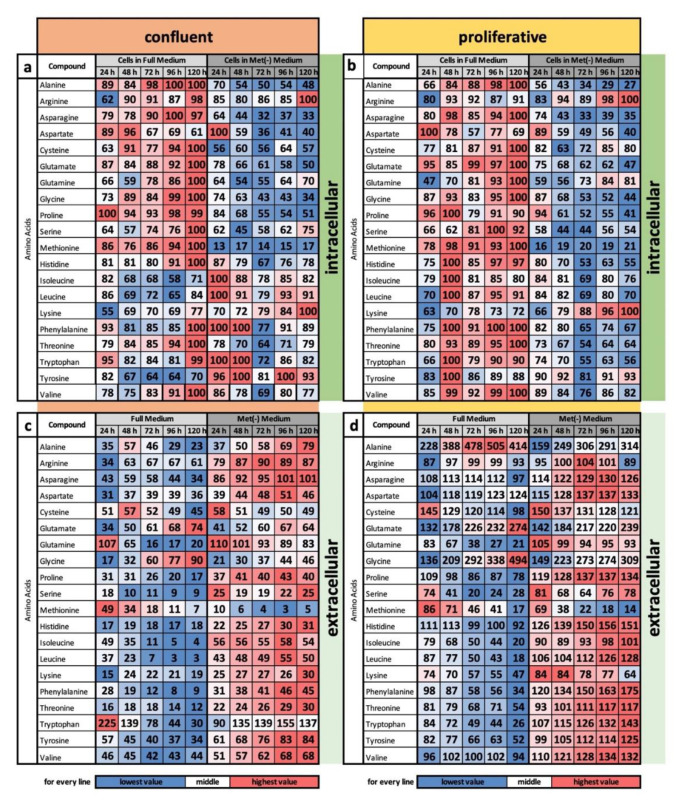
(**a**–**d**) Relative amounts of proteinogenic amino acids under MetR comparing confluent and proliferative conditions with the cell line L929. The metabolism of the murine cell line L929 was analysed in full medium and MetR under both confluent (**a**,**c**) and proliferative conditions (**b**,**d**) over a period of 5 days. For each day of the experiment, the preparation was performed in triplicate. After 24, 48, 72, 96 and 120 h, both the medium (extracellular) and the cell lysates (intracellular) were analysed by LC/MS. The results were reproduced in four independent experiments and finally summarized. In this figure, the results of the proteinogenic amino acids are shown. For the values of the media, the control measurement of the medium used was defined as 100%. For the cell pellets, the highest measured value in each test series was defined as 100%.

**Figure 3 cells-11-00551-f003:**
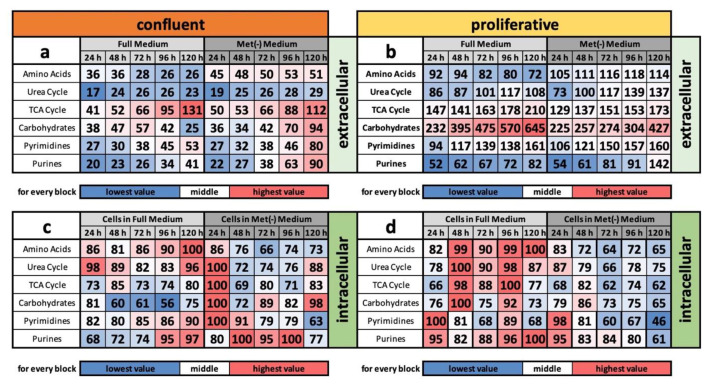
(**a**–**d**) Overview of individual metabolic classes and metabolic groups under MetR in a comparison between confluent and proliferative conditions in the cell line L929. The metabolism of the murine cell line L929 was analysed in full medium and MetR under both confluent (**a**,**c**) and proliferative conditions (**b**,**d**) over a period of 5 days. For each day of the experiment, the preparation was performed in triplicate. After 24, 48, 72, 96 and 120 h, both the medium (extracellular) and the cell lysates (intracellular) were analysed by LC/MS. The results were reproduced in four independent experiments and finally summarised. This figure shows the results of selected classes of substances and metabolic pathways. For the values of the media, the control measurement of the medium used was defined as 100%. For the cell pellets, the highest measured value in each test series was defined as 100%.

**Figure 4 cells-11-00551-f004:**
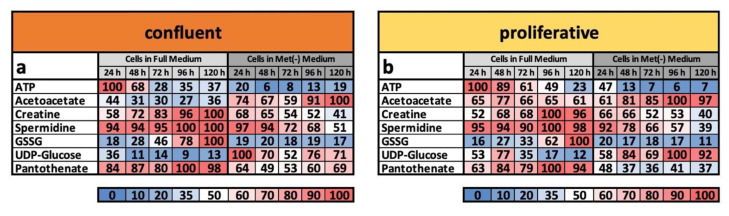
(**a**,**b**) Metabolic footprints of the cell line L929. The metabolism of the murine cell line L929 was analysed in full medium and MetR under both confluent (**a**) and proliferative conditions (**b**) over a period of 5 days. For each day of the experiment, the preparation was performed in triplicate. After 24, 48, 72, 96 and 120 h, the cell lysates (intracellular) were analysed by LC/MS. The results were reproduced in four independent experiments and finally summarised. This figure shows the results of the intracellular metabolites used to define the metabolic footprint for LEM. For the cell pellets, the highest measured value in each test series was defined as 100%.

## Data Availability

An overview of the complete LC/MS results has been added to the Appendix A. The Raw data and results for the two profiles are added as excel files as S1 (proliferative profile) and S2 (non-proliferative profile).

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
