# Peer review of "Low Energy Status under Methionine Restriction Is Essentially Independent of Proliferation or Cell Contact Inhibition"

_cells, 2022, doi:10.3390/cells11030551_

Round 1

Reviewer 1 Report

The article is original, findings are pretty interesting with a huge potential in cancer research. The results are significant for the area and well presented. I enjoyed the reading, it was fluid, precise, and easy to follow.

I have two main concerns of the text, to improve the introduction and discussion:

  1. The authors should explain the origin of the murine cell line. It is clear for them the use of it, however more information regarding the cells origin and the advantages of using it instead of a human cancer cell line. It could be included in the introduction, in a brief paragraph, or the methodology section.
  2. In the discussion, the results are clear, but I wonder about the future implications in the clinic. I strongly suggest justifying the use of murine cells.

Reviewer 2 Report

The manuscript entitled Low Energy Status Under Methionine Restriction is Essentially Independent of Proliferation or Cell Contact Inhibition by Koderer and coauthors studied the effect of contact inhibition on the metabolic profile of L929 cells. The authors found that proliferation and contact inhibition do not affect normal metabolism in L929 cells but under methionine deficiency cells induces low energy metabolism 20 (LEM). The manuscript is straightforward and sound. Here are some questions:

1. Why did the authors choose murine cell line L929 for this study?

2. Results section: Lines 216-218 are not clear. What did the authors want to say here?

3. How methionine regulates amino acids uptake.
